# Synthesis and characterization of a formal 21-electron cobaltocene derivative

Satoshi Takebayashi ®[1] ✉, Jama Ariai ®[2], Urs Gellrich ®[2] ✉,
Sergey V. Kartashov ®[3], Robert R. Fayzullin ®[3] ✉, Hyung-Been Kang[4],
Takeshi Yamane ®[5], Kenji Sugisaki[5,6,7,8] & Kazunobu Sato ®[5]

Metallocenes are highly versatile organometallic compounds. The versatility of the metallocenes stems from their ability to stabilize a wide range of formal electron counts. To date, *d*-block metallocenes with an electron count of up to 20 have been synthesized and utilized in catalysis, sensing, and other fields. However, *d*-block metallocenes with more than formal 20-electron counts have remained elusive. The synthesis and isolation of such complexes are challenging because the metal−carbon bonds in *d*-block metallocenes become weaker with increasing deviation from the stable 18-electron configuration. Here, we report the synthesis, isolation, and characterization of a 21-electron cobaltocene derivative. This discovery is based on the ligand design that allows the coordination of an electron pair donor to a 19-electron cobaltocene derivative while maintaining the cobalt−carbon bonds, a previously unexplored synthetic approach. Furthermore, we elucidate the origin of the stability, redox chemistry, and spin state of the 21-electron complex. This study reveals a synthetic method, structure, chemical bonding, and properties of the 21-electron metallocene derivative that expands our conceptual understanding of *d*-block metallocene chemistry. We expect that this report will open up previously unexplored synthetic possibilities in *d*-block transition metal chemistry, including the fields of catalysis and materials chemistry.

Since the discovery of ferrocene in the 1950s by Kealy and Pauson, as well as Miller et al.[1], and the Nobel Prize-winning work of Fischer and Wilkinson[2,3], various derivatives of metallocenes have been synthesized and have played pivotal roles in important discoveries in a variety of fields, including catalysis[4,5], materials[6–9], energy[10], and medical[11,12] sciences (Fig. 1a). The versatility of metallocenes stems from the ability of the cyclopentadienyl ligand (Cp) and its derivatives to stabilize

metals with a wide range of valence electron counts. To date, *d*-block metallocenes and their derivatives with a formal electron count in the range from 14 to 20, including recently synthesized 16-electron ferrocene dication[13] and 20-electron cobaltocene anion[14], have been isolated. Wilkinson and Hursthouse et al. reported a possible formal 21-electron manganocene derivative, MnCp₂(dmpe) [dmpe: 1,2-bis(dimethylphosphino)ethane] (Fig. 1b)[15]. However, as noted by the

[1]Science and Technology Group, Okinawa Institute of Science and Technology Graduate University, 1919-1 Tancha, Onna-son, Okinawa 904-0495, Japan. [2]Institute of Organic Chemistry, Justus Liebig University Giessen, Heinrich-Buff-Ring 17, Giessen D-35392, Germany. [3]Arbuzov Institute of Organic and Physical Chemistry, FRC Kazan Scientific Center, Russian Academy of Sciences, 8 Arbuzov Street, Kazan 420088, Russian Federation. [4]Engineering Section, Okinawa Institute of Science and Technology Graduate University, 1919-1 Tancha, Onna-son, Okinawa 904-0495, Japan. [5]Department of Chemistry, Graduate School of Science, Osaka Metropolitan University, 3-3-138 Sumiyoshi, Sumiyoshi-ku, Osaka 558-8585, Japan. [6]JST PRESTO, 4-1-8 Honcho, Kawaguchi, Saitama 332-0012, Japan. [7]Present address: Graduate School of Science and Technology, Keio University, 7-1 Shinkawasaki, Saiwai-ku, Kawasaki, Kanagawa 212-0032, Japan. [8]Present address: Quantum Computing Center, Keio University, 3-14-1 Hiyoshi, Kohoku-ku, Yokohama, Kanagawa 223-8522, Japan. ✉e-mail: satoshi.takebayashi@oist.jp; urs.gellrich@org.chemie.uni-giessen.de; robert.fayzullin@gmail.com

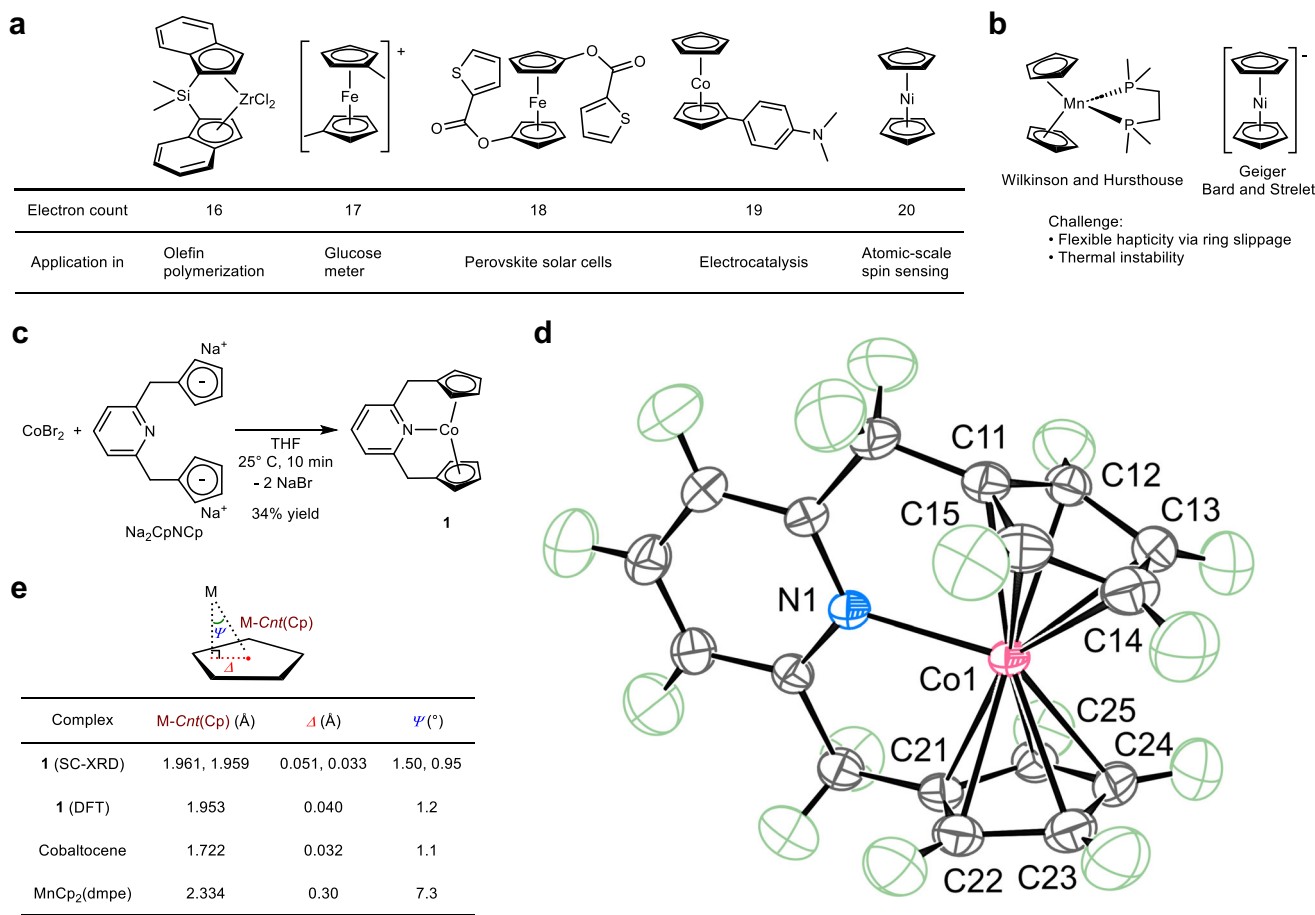

**Fig. 1 | Preparation of 21-electron metallocene derivatives. a** Selected examples of metallocenes and their applications. **b** Previously proposed 21-electron metallocene derivatives. **c** Synthetic route to the 21-electron cobaltocene derivative, complex **1**. THF: tetrahydrofuran. **d** Oak Ridge Thermal Ellipsoid Plot (ORTEP) of **1** at the 80% probability level according to high-resolution single-crystal X-ray diffraction (SC-XRD). Green ellipsoids: hydrogen atoms. **e** Comparison of M−*Cnt*(Cp), *Δ*, and *Ψ* values. M: metal. DFT: density functional theory. dmpe: 1,2-bis(dimethylphosphino)ethane. Cp: cyclopentadienyl.

authors, the formation of this complex does not follow the expected trend of other metallocenes[16]. Furthermore, as with other manganocene complexes, the coordination mode of the Cp ligands in this complex deviates from the ideal $\eta^5$-coordination because of ring slippage[15,17–22]. This anomalous behavior of manganocene complexes is attributed to the primarily electrostatic character of the Mn−C(Cp) interactions, and the observed coordination modes are mainly controlled by the steric factor of the ligands[22–24]. The solid-state zigzag chain structure of MnCp$_2$ is a vivid example of the anomalous coordination chemistry of manganocene complexes[23]. Therefore, according to the available literature data, the formal electron count of MnCp$_2$(dmpe) cannot be conclusively described. Geiger et al.[25] and later Bard and Strelets et al.[26] reported the electrochemical reduction of NiCp$_2$ and proposed the formation of formal 21-electron NiCp$_2^-$ based on cyclic voltammetry (CV) (Fig. 1b). However, since this complex is thermally unstable and cannot be isolated, the possibilities for its characterization are limited.

The 18-electron rule works best for the complexes with strong π-accepting ligands[27] and the existence of formal 14- to 20-electron metallocenes clearly shows that the 18-electron rule is only loosely applicable to metallocene complexes, especially paramagnetic ones. Furthermore, it is worth nothing to point out that a bonding model based on $sd^n$ hybridization and 3c–4e hypervalent bonding[28–30] indicates the possible formation of formal 21-electron metallocene derivatives by exploiting weaker attractive interactions such as higher multi-center donor–acceptor interactions, long-range electrostatics, London dispersion, and/or entropically favorable interstitial vacancies[31]. Nevertheless, the isolation of well-defined *d*-block metallocenes or their derivatives with more than a formal 20-electron count has remained elusive. The difficulty in synthesizing formal 21-electron complexes lies in the fact that increasing deviation from the stable 18-electron configuration destabilizes metal−C(Cp) interactions and promotes the change of hapticity of Cp ligands[32–37] or decomposition of the complexes[25,26]. Here, we report the synthesis, isolation, and detailed theoretical and experimental characterization of a new cobaltocene derivative with a formal electron count of 21 (Fig. 1c).

## Results and Discussion

### Synthesis and spectroscopic characterization

Inspired by the coordination chemistry of Cp$_2$Ni[32], we envisioned that the incorporation of two Cp ligands into a pyridine-based pincer ligand motif might allow the *N*-coordination of the pyridine ligand to the metal center while maintaining the $\eta^5$-coordination mode of the two Cp groups. For this reason, the Na$_2$CpNCp ligand (Fig. 1c)[38] was chosen for the synthesis of metallocene derivatives with formal valence electron counts of more than 20. Prior to this work, the coordination chemistry of *d*-block metals and Na$_2$CpNCp was known only with metals with a formal *d*-electron count of less than four[39–41]. Portionwise addition of Na$_2$CpNCp to a tetrahydrofuran (THF) solution of CoBr$_2$ at room temperature resulted in the formation of a NaBr precipitate. Following the workup, an orange crystalline solid of 2,6-bis(methylenecyclopentadienyl)pyridinecobalt (**1**) was isolated in 34% yield (average of two experiments) (Fig. 1c). Unlike NiCp$_2^-$, complex **1** was

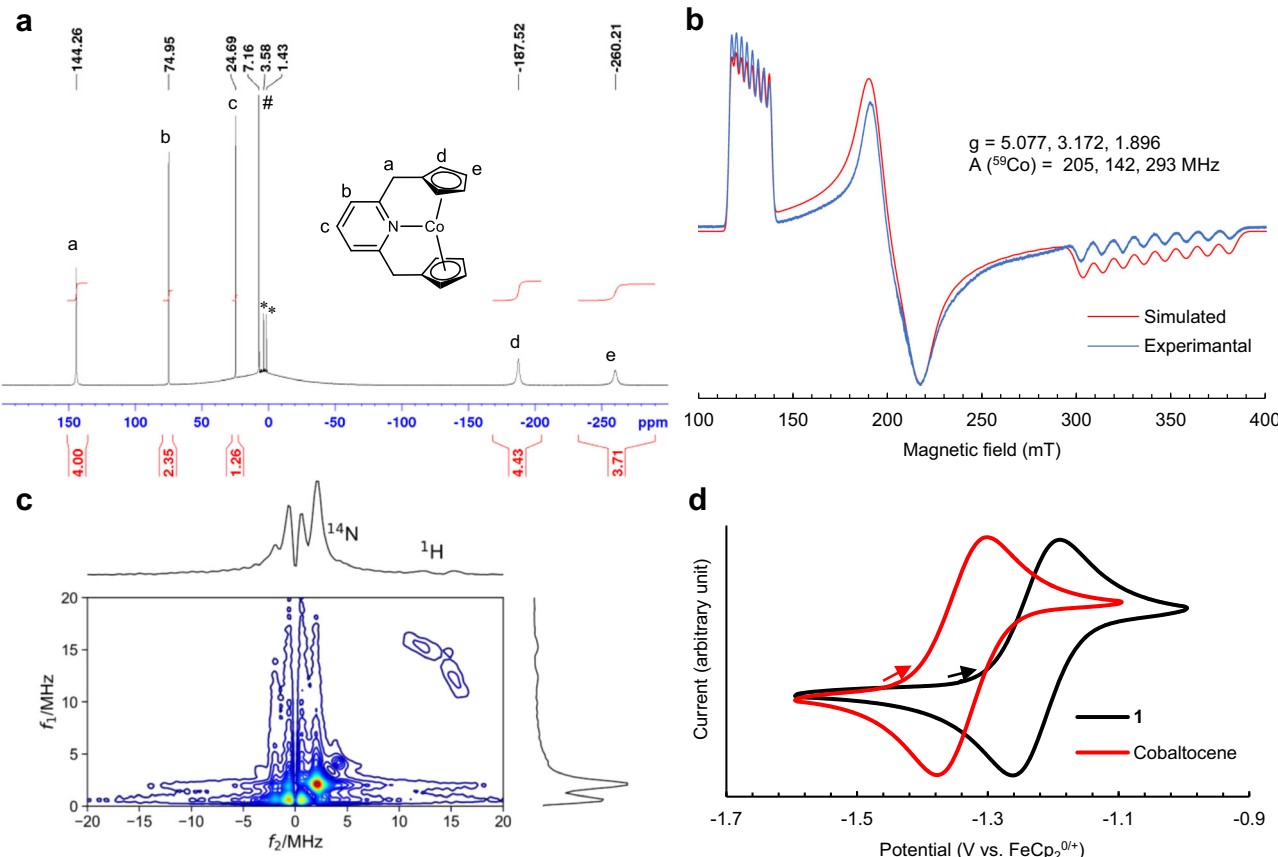

**Fig. 2 | Characterization of complex 1. a** $^1$H NMR (400.15 MHz, 298 K, in $C_6D_6$) spectrum of **1** and signal assignments (a-e). *: signals from residual THF. #: a signal from residual $C_6D_5$H. **b** Experimental (4.2 K, toluene glass) and simulated X-band EPR spectra of **1** using fictitious $S = 1/2$. g: effective g-values. A($^{59}$Co): $^{59}$Co hyperfine coupling constants. **c** Four-pulse HYSCORE spectrum of **1** observed at 320 mT (4 K,

toluene-$d_8$ glass). **d** CV of **1** and cobaltocene recorded with a glassy carbon electrode (3 mm diameter) from −1.6 to −1 V range with a scan rate of 0.01 V s$^{-1}$, in 0.1 M [Bu$_4$N]PF$_6$ THF at 25 °C, and reported vs. FeCp$_2$$^{+/0}$. The starting point (open circuit potential) and direction of scans are indicated by arrows.

stable as a solid or as a solution under nitrogen at ambient temperature.

In order to unambiguously show the bonding situation and formal electron count of **1**, the paramagnetic complex **1** was characterized by a number of methods, including high-resolution single-crystal X-ray diffraction (SC-XRD), $^1$H, $^{13}$C, and $^2$H nuclear magnetic resonance (NMR) spectroscopies, electron paramagnetic resonance (EPR) spectroscopy, vibrating sample magnetometry (VSM), electrospray ionization mass spectrometry (ESI-MS), elemental analysis, and density functional theory (DFT) calculations at the TPSSh-D4/def2-QZVPP// TPSS-D4/def2-TZVPP level of theory. The molecular structure of **1** in the crystal, obtained by precision high-resolution SC-XRD at 100 K with a reciprocal resolution $\sin(\theta_{max}/\lambda)$ of 1.26 Å$^{-1}$ and following multipole refinement[42], supports the *N*-coordination of a pyridine moiety and $\eta^5$-coordination mode of two Cp groups (Fig. 1d). More specifically, the Co-N bond distance of 2.1998(3) Å is within the internuclear Co$^{II}$-N$_{Py}$ distances in the neutral Co$^{II}$PNP pincer complexes, which are in the range from 1.974 to 2.343 Å with an average distance of 2.144 Å (Supplementary Table 2). The Co-C(Cp) distances are in the range from 2.2783(3) to 2.3292(2) Å with an average distance of 2.302 Å, and the Co-Cp centroid (Co-$Cnt$(Cp)) distances are 1.9607(2) and 1.9593(2) Å. These distances are significantly longer than those of cobaltocene (2.112 Å for the average of the libration-corrected Co-C(Cp) distances and ca. 1.722 Å for Co-$Cnt$(Cp))[43] and may indicate a weakening of the Co-C(Cp) interactions. The differences between the maximum and minimum Co-C(Cp) distances of 0.051 and 0.035 Å for the two Cp groups are similar to those of cobaltocene (0.033 Å)[43]. The coordination mode of the Cp groups was further

analyzed by the Cp ring slip parameters $\Delta$ and $\Psi$ (Fig. 1e)[44]. The $\Delta$ and $\Psi$ values of complex **1** are among the smallest reported for $\Delta$ and $\Psi$[35,44] and strongly support the $\eta^5$-coordination mode of the two Cp groups. In comparison, the proposed 21-electron manganocene derivative, MnCp$_2$(dmpe), has a significantly larger deviation from the ideal $\eta^5$-coordination mode (Fig. 1e). The $\Delta$ and $\Psi$ values derived from the gas-phase DFT calculations are in agreement with the experimental values, supporting that crystal packing effects are not essential for the $\eta^5$-coordination (Fig. 1e). The DFT optimized structure reproduces the equidistance of the Co-C(Cp) bonds with an average distance of 2.297(21) Å and a short Co-N distance of 2.2176 Å. Furthermore, the average Wiberg bond index for the Co-C(Cp) bonds of 0.15(2) is similar to that of cobaltocene (0.22(5)) as calculated at the same level of theory, and a practically uniform distribution of negative charges on the Cp rings was observed by the calculations (Supplementary Table 3).

In solution, the $^1$H NMR spectrum of **1** showed five signals between −260 and +144 ppm (Fig. 2a). All the $^1$H NMR signals were assigned based on selective deuteration of the Cp groups and two-dimensional NMR experiments (Supplementary Figs. 6, 7, and 12). The two broader signals at the higher field are from the Cp rings, and the three sharper signals at the lower field are from the pyridine ring and CH$_2$ groups of the CpNCp ligand. The $^{13}$C NMR spectrum shows four signals between −504 and +410 ppm (Supplementary Fig. 5), and the signals are partially assigned based on two-dimensional NMR and the splitting pattern of the signals. Three $^{13}$C NMR signals are missing, most likely due to the close proximity of the Cp groups to the paramagnetic Co center. Magnetic measurements conducted by the Evans NMR method (298 K,

in $C_6D_6$) and VSM (298 K, bulk solid) showed that **1** has an effective magnetic moment $\mu_{eff}$ of 4.3 and 3.99 $\mu_B$ ($\mu_B$: Bohr magneton) in solution and solid state, respectively, which indicates that **1** has the $S = 3/2$ ground electron spin state. The $S = 3/2$ ground electron spin state was further supported by continuous wave (cw)-EPR, where a large deviation of effective g-values from that of a free electron was observed due to the sizable zero-field splitting of the $S = 3/2$ complex (Fig. 2b). The EPR spectrum also clearly shows the presence of large $^{59}$Co hyperfine splitting, which supports the presence of a cobalt-centered radical.

The presence of the Co–N bond in the frozen solution was supported by pulse-EPR measurements, which observed the electron spin echo envelope modulation (ESEEM) effects[45]. The two-dimensional field-swept two-pulse and three-pulse ESEEM spectra observed at 4 K (Supplementary Figs. 30 and 31) showed the nuclear modulation effects due to the N atom coupled with the Co atom, indicating the presence of bonding interaction between them. Furthermore, we applied four-pulse hyperfine sublevel correlation (HYSCORE) spectroscopy[46] to measure the nuclear spins coupled with the Co atom. The HYSCORE spectrum (Fig. 2c) showed the correlation peaks due to the $^1$H and $^{14}$N atoms. The $^1$H-HYSCORE signals spreading over a frequency range of 3–4 MHz in the angular selective spectra were assigned to the protons belonging to the Cp groups. Based on the cw-EPR spectrum and the modulation effects, we determined spin-Hamiltonian parameters of **1** with $S = 3/2$[47] (Supplementary Fig. 33, Supplementary Table 1) as well as fictitious $S = 1/2$ (Fig. 2b). The experimental tensors showed that the principal axis with the largest principal value of the $^{14}$N-hyperfine coupling (A) tensor coincides with that of the $^{59}$Co-A tensor. This observation indicates the presence of electronic interaction between the Co and N atoms in the frozen solution.

The cyclic voltammogram of **1** showed reversible $Co^{3+/2+}$ half-wave potential at –1.23 V (vs. $FeCp_2^{+/0}$ in MeCN), which is 0.11 V higher or less negative than that of cobaltocene (Fig. 2d). This change in the $Co^{3+/2+}$ half-wave potential is not due to the introduction of the pyridinylmethyl group on each Cp ring, since such a substitution is expected to decrease the reduction potential by about 0.02 V[48]. Thus, the *N*-coordination of the pyridine group unexpectedly increases the reduction potential of **1** despite the increase in formal electron count.

## Examination of the interatomic interactions

The bonding situation in **1** was further investigated by the quantum topological analysis of the experimental electron density $\rho(\mathbf{r})$[49] and electrostatic potential $\varphi_{es}(\mathbf{r})$[50] in the crystal. The analysis identified six bond critical points (BCPs) with negative near-zero values of the electronic energy density $h(\mathbf{r})$ corresponding to the Co–N and Co–C(Cp) coordinate bonds (Supplementary Fig. 34)[51]. In comparison, the analysis of $\rho(\mathbf{r})$ derived from DFT calculations resulted in a different set of Co–C(Cp) coordinate bonds (Supplementary Figs. 34 and 35), while Co–N was still present. It is known that the location of BCPs alone is not sufficient to correctly describe metal···Cp hapticity[52]. Therefore, the distribution of the Laplacian of electron density $\nabla^2\rho(\mathbf{r})$ (Fig. 3a and Supplementary Fig. 36) was studied. Examination of the $\nabla^2\rho(\mathbf{r})$ maps reveals that the charge concentration lobes associated with the N atom and each C atom of the Cp rings (lobes of red contour in Fig. 3a) are directed toward the Co atom in the electron depletion region (blue contours) approximately along the internuclear lines. Moreover, the N-atom lobe is directed into the furcation region between the valence shell charge concentrations of the Co atom, supporting the donor–acceptor nature of the coordinate Co–N bond. These observations suggest the $\eta^5$-coordination of both Cp groups as well as the *N*-coordination of the pyridine group. The experimental and theoretical values of $\rho(\mathbf{r})$, $\nabla^2\rho(\mathbf{r})$, and $h(\mathbf{r})$ at the BCP for Co–N: 0.057, 0.220, and –0.006 a.u. and 0.055, 0.218, and –0.011 a.u., respectively, showed good agreement. An approximate Co–N delocalization index

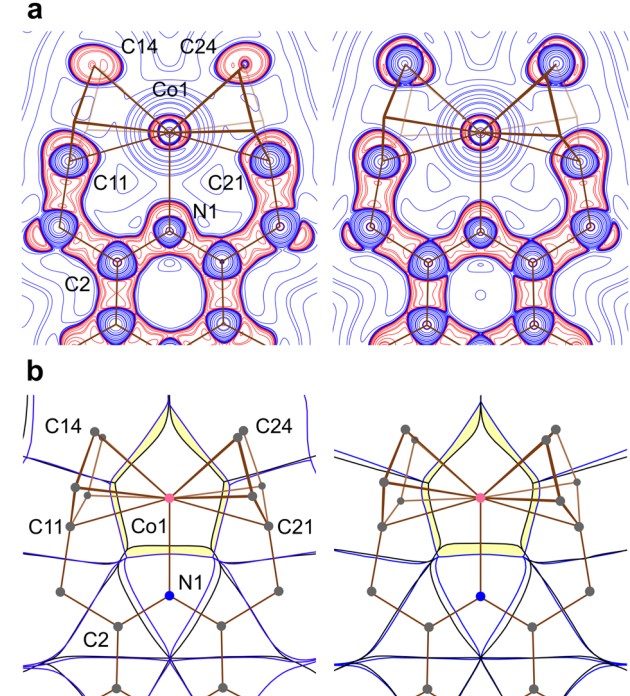

**Fig. 3 | Quantum chemical topology of 1.** Experimental maps based on SC-XRD data (left) and theoretical maps based on gas-phase DFT calculations (right) are compared. Map planes pass through the atoms Co1, N1, and C2. **a** Contour maps of $\nabla^2\rho(\mathbf{r})$; the logarithmic scale in the form of $\pm 1, 2, 4, 8 \cdot 10^n$ ($-2 \leq n \leq 3$) e Å$^{-5}$ is adopted; blue and red colors correspond to positive and negative function values, respectively. **b** Superposition of the zero-flux boundaries of $\rho$-basins (black) and $\varphi_{es}$-basins (blue) in the gradient vector fields $\nabla\rho(\mathbf{r})$ and $\nabla\varphi_{es}(\mathbf{r})$.

(DI) of 0.78 and an average Co–C(Cp) DI of 0.52(5), obtained from DFT calculations, further support the presence of interactions between Co and N, as well as all C(Cp) atoms[52,53].

The Coulombic attraction between Co and N as well as C(Cp) atoms was examined by superposing trajectory maps of the gradient vector fields of charge density and electrostatic potential, $\nabla\rho(\mathbf{r})$ and $\nabla\varphi_{es}(\mathbf{r})$ (Fig. 3b)[50,54]. The electrons within the volumetric overlapping area of $\nabla\rho(\mathbf{r})$ and $\nabla\varphi_{es}(\mathbf{r})$ (colored yellow in Fig. 3b) belong to the atomic $\rho$-basins of N and all C(Cp) but simultaneously fall into the neighboring pseudoatomic $\varphi_{es}$-basin of Co; therefore, these captured electrons are drawn toward the Co nucleus by the electrostatic force $\mathbf{F}_{es}(\mathbf{r}) = \nabla\varphi_{es}(\mathbf{r})$. This area represents the trace of the expansion of the ligand atoms of the first coordination sphere, capturing electrons, and the compression of the Co atom losing electrons as a result of the chemical bond formation starting from free neutral atoms. The electron transfer thus defined is estimated to be 0.90 and 0.98 e for **1** in the crystal according to experimental diffraction data and for the free molecule from DFT calculations, respectively. The action of the electric field generated by the cobalt atom inside the nearest 11 atoms of the ligand, as well as the joint total interatomic charge transfer from Co to N and C(Cp), indicates that all 10 C(Cp), as well as the N-atom, form a "ligand binding field" around the Co-coordination center. The experimental and computational studies strongly support the presence of the Co–N bond and two $\eta^5$-coordinated Cp ligands, and together show that complex **1** is a formal 21-electron cobaltocene derivative.

## Attempted synthesis of 22-, 20-, and 19-electron analogs

Aiming at the formation of a 22-electron analog, we attempted to synthesize the analogous Ni(CpNCp) complex. However, the reaction between $Na_2CpNCp$ and $NiCl_2(DME)$ (DME: 1,2-dimethoxyethane) resulted in the formation of a dimeric nickel complex **2** that lacked

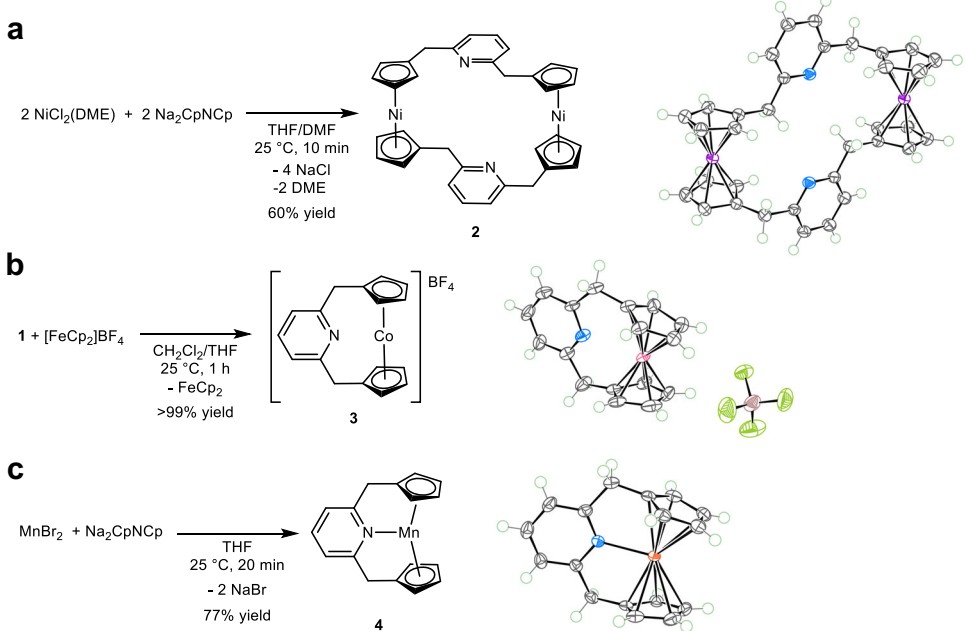

**Fig. 4 | Attempted formation of formal 22-, 20-, and 19-electron analogs.**
**a** Synthetic route to the dimeric Ni$^{II}$ complex **2**. THF: tetrahydrofuran. DMF:
dimethylformamide. DME: 1,2-dimethoxyethane. **b** Oxidation of **1** with [FeCp$_2$]BF$_4$
to generate Co$^{III}$ complex **3**. **c** Synthetic route to the Mn$^{II}$ complex **4**. Insert: Mole-
cular structure of corresponding complexes at the 80 % probability level for non-
hydrogen atoms according to SC-XRD.

Ni−N bonds (Fig. 4a). Therefore, the addition of one more electron
cancels out the stability of the complex gained by the formation of the
metal−N bond. The one-electron oxidation of **1** was also investigated to
elucidate if a formal 20-electron analog could be prepared. Oxidation
of **1** by [FeCp$_2$]BF$_4$ resulted in the quantitative formation of a dia-
magnetic cobaltocenium complex **3** (Fig. 4b). The SC-XRD analysis of **3**
revealed long Co⋯N internuclear distances of 2.745(4) and 2.790(5) Å,
which vary significantly for two molecules in the asymmetric cell.
Furthermore, the average Co−C(Cp) bond length is significantly
shortened from 2.302 to 2.084 Å and close to that of [CoCp$_2$]BF$_4$
(2.041 Å)[55]. The variable temperature (from −40 to 80 °C) NMR mea-
surements showed no sign of reversible coordination of the pyridine
group, and the $^{15}$N NMR chemical shift of **3** is close to that of
Na$_2$CpNCp. Therefore, **3** is best described as a formal 18-electron
complex without Co−N bonding. The absence of Co−N bonding is
likely due to the formation of a stronger attractive interaction between
the anionic Cp groups and the Co$^{3+}$ center, as evidenced by the
shortening of the Co−C(Cp) bonds, which increases the energy penalty
required to form a Co−N bond. To further investigate the coordination
chemistry of the CpNCp ligand, we synthesized the Mn complex **4**
(Fig. 4c). A μ$_{eff}$ of 6.2 (in C$_6$D$_6$, at 298 K) and the cw-EPR spectrum and
its simulation with S = 5/2 spin state supports an S = 5/2 ground elec-
tron spin state of **4** (Supplementary Fig. 27). The SC-XRD analysis of **4**
showed that the Mn−N distance is 2.2709(14) Å, which is within the Mn
−N distance of other N-donor-ligated manganocene derivatives[24]. The
Cp ring slip parameters $\Delta$ of 0.213 and 0.105 Å as well as $\Psi$ of 5.7° and
2.8° are relatively small but significantly larger than those of **1** (Fig. 1e).
Thus, like other manganocene derivatives, the coordination chemistry
of **4** is more strongly influenced by the ligand steric factor[24]. These
studies of the Ni, Co, and Mn complexes led us to further investigate
the origin of the stability of complex **1** and the instability of the
metal−N bonds in complexes **2** and **3** by computational methods.

## Origin of the stability
The origin of the stability of **1** and the unexpected increase in the
Co$^{3+/2+}$ reduction potential were investigated by computational
methods. Using DFT calculations, we were able to locate an isomer of

**1** with an S = 1/2 spin state in which the coordinate Co−N bond is
broken, as a local minimum energy structure (**1′**, Fig. 5).

The relaxed potential energy surface scans of the Co−N distance
reveal that upon decreasing the Co−N distance, the 21-electron con-
figuration with a spin state of S = 3/2 is more stabilized than the 19-
electron configuration with S = 1/2 (Supplementary Fig. 41), with **1**
being 5 kcal mol$^{-1}$ (Gibbs free energy at 298 K) more stable than **1′**.
Thus, the coordination of nitrogen significantly stabilizes complex **1**.
Figure 5 shows the frontier molecular orbital (MO) diagrams of
cobaltocene, **1′**, and **1**, derived canonically from Kohn−Sham quasi-
restricted orbitals (KS-QROs)[56]. As expected, the MO diagrams of
cobaltocene and **1′** are similar, and no significant change in the energy
of the singly occupied molecular orbital (SOMO) was observed.
Examination of the canonical KS-QROs of **1** revealed that the Co−N
bond in **1** results from the σ-type interaction between the metal d-
orbital and the nitrogen lone pair. The formation of the Co−N bond has
two consequences. First, it generates the Co−N antibonding orbital,
which is singly occupied (Fig. 5, SOMO1). Thus, the Co−N bond for-
mally has a bond order of 0.5. This situation is reminiscent of the
formation of 19-electron adducts from the 17-electron complexes,
where the formation of one-half of a metal-ligand bond stabilizes the
19-electron complex[57]. Second, the formation of the Co−N bond sig-
nificantly lowers the energy level of the SOMO and the lowest unoc-
cupied molecular orbital (LUMO) of **1′**, and both can be singly occupied
to form an S = 3/2 ground electron spin state (Fig. 5, SOMO2 and
SOMO3). These two orbitals are Co⋯Cp antibonding, and their partial
occupation explains the significant elongation of the Co−C(Cp) inter-
nuclear distances in **1**. The decrease in the energy of the highest
SOMO3 explains the observed increase in the Co$^{3+/2+}$ reduction
potential. These half-filled Co⋯Cp antibonding orbitals allow the Cp
ligands to maintain the $\eta^5$-coordination mode. Therefore, the stability
of **1** is a product of the formation of one-half of a Co−N bond and the
half-filled two Co⋯Cp antibonding orbitals, and the pincer ligand motif
that holds all these weak Co−N and Co−C(Cp) bonds together. Indeed,
a control experiment showed that coordination of pyridine to cobal-
tocene does not occur without the chelate effect provided by the
pincer motif (see the Supplememntary Information for details). This

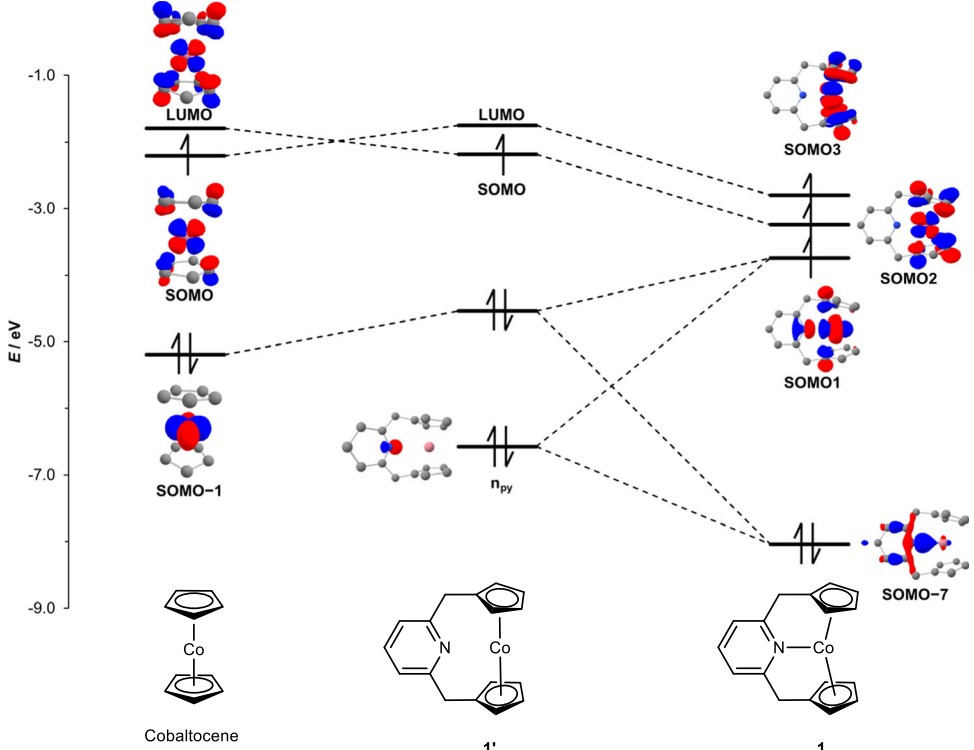

**Fig. 5 | Comparison of MO diagram of cobaltocene, 1', and 1.** Inset: canonical KS-QROs (isovalues at 0.05-0.13 a.u.), blue lobes: positive phases, red lobes: negative phases. For a detailed MO diagram and KS-QROs, please see Supplementary Figs. 43-46. LUMO: lowest unoccupied molecular orbital. SOMO: singly occupied molecular orbital. SOMO−1: a molecular orbital at one energy level below SOMO. SOMO−7: a molecular orbital at seven energy level below SOMO1. $n_{py}$: nitrogen lone pair electrons.

type of complex stabilization by half-filled metal−ligand antibonding orbitals contrasts with the way formal 20-electron $M(CO)_8$ (M = Zr, Hf)[58] and $[M(CO)_8]^-$ (M = Sc, Y, La)[59] species are stabilized by the presence of a filled purely ligand based MO. Our MO analysis also qualitatively explains the formation of **2** and **3**. The hypothetical formal 22-electron Ni(CpNCp) complex should have a fully occupied Ni−N antibonding orbital, leading to the cleavage of a Ni−N bond and the formation of **2**. The removal of one electron from **1** should lead to the pairing of the remaining two unpaired electrons in **1** to achieve a more stable low-spin configuration, in which the Co−N bond does not exist.

In summary, we have synthesized the formal 21-electron cobaltocene derivative **1** by utilizing the chelate effect of the CpNCp pincer ligand, which enables the coordination of the nitrogen lone pair donor while maintaining the $\eta^5$-coordination mode of two Cp groups. The structural analysis, quantum-chemical topological studies, and DFT calculations strongly support the presence of the Co−N bond and two $\eta^5$-coordinated Cp ligands, and thus the formation of the 21-electron cobaltocene derivative. The stability and unexpected redox property of the 21-electron complex **1** originate from the half-filled Co−N and Co···Cp antibonding orbitals, and the chelation effect of the pincer ligand. The formal 21-electron metallocene presented here expands our understanding and perception of metallocene and organometallic chemistry, as well as chemical bond theory. It is anticipated that this discovery will open up unexplored possibilities in transition metal chemistry, especially in the field of catalysis and materials science.

## Methods

### General considerations and materials for the synthetic study

All reactions were carried out under an $N_2$ atmosphere using an MBRAUN glovebox, UNILAB Plus SP, equipped with an MB-20-G gas purifier, an MB-LMF-2/40-REG regenerable solvent trap, and an MB-GS-35 -35 °C freezer. All glasswares were dried overnight at 170 °C and

cooled down under vacuum in the glovebox antechamber. All solvents were reagent grade or higher. *n*-Pentane (≥ 99.0%), hexane (≥ 95.0%, *n*-hexane with minor amounts of isomers of methylpentane and methylcyclopentane), *n*-heptane (≥ 99.0%), dichloromethane (≥ 99.5%), benzene (≥ 99.7%), toluene (≥ 99.8%), acetonitrile (≥ 99.8%), methanol (≥ 99.8%), tetrahydrofuran (Sigma-Aldrich 401757-1 L, anhydrous, ≥99.9%, inhibitor-free), diethyl ether (≥ 99.8%, inhibitor-free), hexamethyldisiloxane (Sigma-Aldrich 205389-500 ML, ≥98%), and dimethylformamide (DMF, ≥99.0%) were dried over MS3A (dried overnight in a 200 °C oven and cooled overnight under vacuum in the glovebox antechamber) in the glovebox for more than 2 days and stored in the glovebox. Common chemicals were purchased, kept in the glovebox, and used as received unless stated otherwise. Anhydrous $CoBr_2$ (green powder, >97%) was purchased from Fujifilm Wako Chemicals. $NiCl_2(DME)$ (yellow powder, 98%) was purchased from Sigma Aldrich. $MnBr_2$ (pink powder, 99%) was purchased from Acros Organics. Celite filtration was carried out using a pipet, cotton wool, and Celite®545, which was dried overnight in a 170 °C oven, cooled down overnight under vacuum in the glovebox antechamber, and kept in the glovebox. The dimensions of the 20 mL vial are 60 mm in height, 28 mm in outer diameter, and the Teflon-coated stirring bar is 15 mm long and 5 mm in diameter.

### Instrumental analysis methods

**NMR spectroscopy.** All deuterated solvents were purchased and dried over MS3A in the glovebox for more than 2 days and kept in the glovebox. NMR spectra were recorded using a Bruker Avance III-400N spectrometer and an Avance III NEO 500 spectrometer equipped with a cryoprobe. $^1H$ and $^{13}C$ NMR chemical shifts are reported in parts per million (δ) relative to TMS (0 ppm) with the residual solvent signal ($CDCl_3$: 7.26 ($^1H$) and 77.16 ($^{13}C$) ppm, $C_6D_6$: 7.16 ($^1H$) and 128.06 ($^{13}C$) ppm, THF-$d_8$: 1.72 ($^1H$) and 25.31 ($^{13}C$) ppm, toluene-$d_8$: 2.08 ($^1H$) and

20.43 ($^{13}$C) ppm, DMSO-$d_6$: 2.50 ($^1$H) and 39.52 ($^{13}$C) ppm, methanol-$d_4$: 3.31 ($^1$H) and 49.00 ($^{13}$C) ppm, CD$_3$CN: 1.94 ($^1$H) and 118.26 ($^{13}$C) ppm) as the internal references $^2$H NMR chemical shifts are reported in parts per million ($\delta$) relative to residual solvent signal as the internal reference. $^{11}$B, $^{15}$N, and $^{19}$F NMR chemical shifts are reported in parts per million ($\delta$) relative to BF$_3$·OEt$_2$ in CDCl$_3$ (0 ppm), NH$_3$(liquid) (0 ppm), and CFCl$_3$ (0 ppm), respectively, as external references.

NMR peak assignments of diamagnetic compounds were made using $^1$H-$^1$H-gCOSY, $^1$H-$^{13}$C-HSQC, and $^1$H-$^{13}$C-HMBC NMR experiments. Abbreviations for NMR spectra are s (singlet), d (doublet), t (triplet), q (quartet), quint (quintet), sep (septet), dd (doublet of doublet), td (doublet of triplet), dq (doublet of quartet), m (multiplet), and br (broad). $^1$H NMR signals of paramagnetic complexes are reported with chemical shift ($\delta$) and line width at half-height ($\Delta v^{1/2}$). Air-sensitive NMR samples were prepared in the nitrogen glovebox using a J. Young NMR tube or a standard NMR tube sealed with a septa and parafilm.

**NMR measurement of paramagnetic complexes.** $^1$H NMR of paramagnetic complexes (typically 20 mM solution) was measured with delay time (d1) of 0.1 s, time domain data points (td) of 128k, and scan number (ns) of 1 to 128, and spectral width (sw) of 200 to 600 ppm. Proton-coupled $^{13}$C NMR experiments were carried out using an Avance III NEO 500 spectrometer equipped with a cryoprobe. For proton coupled $^{13}$C NMR experiments, a delay time (d1) of 0.1 s, a time domain data points (td) of 64k, a scan number (ns) of 64k, and a spectral width (sw) of 1100 ppm were used.

**Measurement of effective magnetic moment by Evans' method.** Measurement of the effective magnetic moment by Evans' method was carried out using glass capillaries containing C$_6$D$_6$ (measurement in C$_6$D$_6$) as external standards.

**Single-crystal X-ray diffraction (SC-XRD).** The X-ray diffraction experiments for **1-3** were performed on a Bruker D8 Venture diffractometer equipped with a PHOTON II CPAD detector and an I$\mu$S 3.0 microfocus X-ray source (Mo $K\alpha$ radiation). The X-ray diffraction data for the single crystal **4** were collected on a Rigaku XtaLab PRO instrument equipped with a PILATUS3 R 200 K hybrid pixel array detector and a MicroMax$^{TM}$-003 microfocus X-ray tube (Mo $K\alpha$). Data were collected at 100 K according to recommended strategies, then processed and corrected. All structures were solved using SHELXT[60]. Structures **2-4** were refined by the full-matrix least-squares using SHELXL[61]. Non-hydrogen atoms were refined anisotropically.

**VSM measurement.** The VSM measurement was carried out using the Quantum Design PPMS DynaCool VSM module. VSM powder sample holders (part #: 4096-388) were weighed outside the glovebox using a microbalance and placed inside in the glovebox. All powdered samples were packed and sealed in the sample holders in the nitrogen glovebox. The sealed samples were weighed outside the glovebox using microbalance to calculate the weight of the samples.

**FTIR spectroscopy.** IR spectra were recorded using a Nicolet iS5 FT-IR spectrophotometer and are reported in absorption frequency (cm$^{-1}$). Abbreviations for FT-IR spectra are s (strong), m (medium), and w (weak).

**High-resolution mass spectrometry (HRMS).** HRMS data were recorded on a Thermo Scientific LTQ-Orbitrap mass spectrometer, using electrospray ionization (ESI) mode.

**Elemental analyses.** Elemental analyses were conducted using an Exeter Analytical CE-440 elemental analyzer. Empty tin cups were weighed outside the glovebox using a microbalance and brought in the glovebox. All samples were weighed and sealed in the tin cups in the nitrogen glovebox. The sealed samples were weighed outside the glovebox using microbalance to calculate weight of the samples. The N$_2$ gas in the tin cups was replaced by argon through three vacuum-argon refill cycles. All the samples were analyzed using an autosampler under a He atmosphere.

**Cyclic voltammetry (CV).** CV was measured inside the nitrogen glovebox using an ECstat-301WL potentiostat equipped with a glassy carbon working electrode (3.0 mm diameter), a platinum wire counter electrode (0.5 mm diameter), and an Ag/Ag$^+$ reference electrode (Ag wire in 0.01 M AgNO$_3$ in 0.1 M [Bu$_4$N]PF$_6$ MeCN solution). The sample solution was prepared by dissolving an appropriate sample in 0.1 M [Bu$_4$N]NPF$_6$ THF solution. All potentials are reported using the FeCp$_2^{0/+}$ couple (0 mV) as an external reference.

### Computational Methods

DFT computations were performed with the ORCA software package (versions 5.0.0, 5.0.1, 5.0.3) unless otherwise noted[62–64]. For structure optimizations and harmonic vibrational frequencies, we employed the TPSS meta-GGA functional[65–67]. Additionally, we utilized Grimme's latest additive dispersion correction D4[68,69] and an Ahlrichs-type basis set with triple-zeta quality (def2-TZVPP)[70]. Minima on the potential energy surface were confirmed by the absence of imaginary frequencies. For accurate electronic energies and quasi-restricted orbitals (QROs)[56] we used the hybrid variant of this functional, i.e., TPSSh, together with the quadruple-zeta quality basis set (def2-QZVPP)[70]. We tested the influence of relativistic effects by incorporating the zeroth-order regular approximation (ZORA) at the scalar spin-free level of the theory[71]. As recommended, these calculations use a relativistically recontracted basis set (ZORA-def2-$X$VP) and a decontracted auxiliary basis set (SARC/J)[72]. Unless noted otherwise, the resolution of identity (RI) and chain-of-spheres (RIJCOSX) approximations, as implemented in ORCA, were used for (meta)-GGA and hybrid functionals, respectively, together with the appropriate auxiliary basis set (def2/J)[72,73]. All computations performed with ORCA utilized the tight SCF convergence criteria and the default integration grid (defgrid2).

Wiberg bond indices were determined in the framework of natural bond orbital (NBO 7.0.10) analysis[74]. The topological analysis of the theoretical electron density $\rho(\mathbf{r})$ and electrostatic potential $\varphi_{es}(\mathbf{r})$ was performed with Multiwfn 3.8(dev)[75]. For this purpose, we carried out single-point energy computations at TPSSh/def2-QZVPP level using the Gaussian16 software package (revision C.01) to generate a Gaussian checkpoint file as input (tight SCF convergence criteria, ultrafine integration grid)[76]. The electrostatic potential $\varphi_{es}(\mathbf{r})$ was evaluated using the built-in code[77]. The protocol for generating superposition maps of the zero-flux surfaces defined in the gradient vector fields $\nabla\rho(\mathbf{r})$ and $\nabla\varphi_{es}(\mathbf{r})$ was described in detail earlier[78].

### Quantum crystallography

The X-ray diffraction data of **1** were collected at 100 K with a reciprocal resolution $\sin(\theta_{max}/\lambda)$ of 1.26 Å$^{-1}$. The multipole refinement of **1** was performed within the Hansen−Coppens formalism as implemented in MoPro[79]. The anharmonic atomic motion of Co was modeled using the Gram−Charlier expansion of the temperature factors. A block refinement of the charge density parameters and the Gram−Charlier coefficients was applied analogously to the procedure described earlier[80].

The analysis of the multipole-derived electron density $\rho(\mathbf{r})$ and electrostatic potential $\varphi_{es}(\mathbf{r})$ was performed in WinXPRO[81] analogous to the published procedures[82]. The energy density $h(\mathbf{r})$ was approximated according to Kirzhnits[83].

## Data availability

All data are available in the main text or the supplementary information. Additional data are available from the corresponding authors upon request. Crystallographic data for the structures reported in this

article and its supplementary information were deposited at the Cambridge Crystallographic Data Centre under deposition numbers CCDC 2220152 (**1**), 2220153 (**1**, multipole model), 2220154 (**2**), 2220155 (**3**), and 2220156 (**4**). Copies of the data can be obtained free of charge via https://www.ccdc.cam.ac.uk/structures/.

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

## Acknowledgements

Funding for this work was provided by the Okinawa Institute of Science and Technology Graduate University (OIST), the Japan Society for the Promotion of Science (22K05134 (S.T.), 19H05621 (K.Sa.)), the German Research Foundation (DFG, GE_3117-1-1 (U.G.)), and the Russian Science Foundation (grant No. 21-73-10191 in part of the quantum topological study (S.V.K. and R.R.F.)). This work was supported by the OIST instrumental analysis and engineering sections.

## Author contributions

S.T. designed the study, organized collaborations, synthesized and characterized complexes, conducted cw-EPR and SC-XRD measurements, and prepared VSM samples. J.A. and U.G. conducted DFT calculations for structure optimization study, MO analysis, and chemical topology study. R.R.F. and S.V.K. analyzed SC-XRD data and conducted quantum crystallography and chemical topology studies. H.-B.K. conducted VSM measurements. T.Y. and K.Sa. conducted the EPR study. K.Su. conducted DFT calculations for the EPR study. For the preparation of manuscript, R.R.F. wrote the part describing the quantum topological study, and K.Sa. wrote the part describing the pulse-EPR study. S.T. combined these parts and wrote the original manuscript. S.T., J.A., U.G., R.R.F., K.Sa., and H.-B.K. revised and edited the original manuscript.

## Competing interests

The authors declare no competing interests.
