## [Peer Review File · Nature Communications]

Synthesis and characterization of a formal 21-electron cobaltocene derivativeReviewers' Comments:

Reviewer #1:

Remarks to the Author:

This manuscript presents a comprehensive investigation of a 21-electron cobalt complex stabilized by a constrained geometry configuration ligand. The authors have extensively characterized the compound, performed detailed theoretical calculations, and studied its reduction reactions to confirm the presence of Co-N bonds, thus proving the 21-electron structure of the central metal. The characterization analysis is convincing, and the results are sufficiently innovative. However, I have some reservations regarding the compound's significance as it may be an isolated case. Therefore, I recommend this manuscript for publication in Nat. Comm. after the following issues are properly addressed:

The authors mention their unsuccessful attempt to synthesize a similar complex with a Ni center. Have they explored Fe and Mn as well? It would be valuable to provide experimental information demonstrating that this ligand design strategy is not limited to cobalt, considering the relative ease of such synthesis attempts.

The authors propose that the compound's stability arises from the lower energy of the three half-filled orbitals formed by N coordination. However, this statement raises questions, as it implies that cobaltocene in pyridine solution would exhibit the same electronic structure if energetically favorable.

Reviewer #2:

Remarks to the Author:

First, I must note that I am a synthetic chemist with an interest in the use of stable metallocenes (ferrocene..) for ligand synthesis/catalysis. Thus many of the details and techniques used in this manuscript are outside of my area of expertise. This notwithstanding, it appears that the work is thorough and very well done. I'm assessing this principally on the novelty of the work, and in this context the claim for a formal 21-electron metallocene derivative is significant, and to my (perhaps limited) knowledge is established beyond doubt. Of particular note is the oxidised cobaltocenium compound **3** which does not contain the C-N bond (i.e. just 18 electron), and the calculations comparing **1'** with **1** and the experimental verification of the 3/2 spin state. I'm less convinced about the possible impact of what appears to be an organometallic oddity on the wider fields of catalysis and materials, but that is for the future, and on the basis of the novelty of the result I am happy to recommend publication.

Minor point/suggestions.

Include solvent label in the ¹H NMR (in addition to THF)?

As THF is present, is the microanalysis data correct?

Figure 5. Change orbital representations so they are aligned the same as the structures (pyridine left).

Reviewer #3:

Remarks to the Author:

The authors present a new metallocene that for the first time has a nominal electron count as high as 21, significantly exceeding the known 18-electron rule. That such a bonding situation could be stabilized is remarkable, since the additional electrons occupy antibonding states. To achieve this, they used a pyridine-based pincer ligand that provides an additional coordinating N atom. Given the authors' experimental results on their attempts to make similar complexes and the very specific electronic situation they uncovered through DFT calculations (competition to N not coordinating), it is not very likely that the concept can be applied to many other cases. Nonetheless, this is an important contribution to the largely established, yet still explored, field of metallocenes. The used combination of experimental techniques and quantum chemical calculations provides a sound picture of the nature of the new compound. The manuscript was developed with care and has a

high formal standard. Still the manuscript could be improved in few aspects.

(1) Can the results of the extended bonding analysis be condensed into a (simplified) picture of chemical bonding that can be adopted in textbooks? Where are the electrons? What are the calculated partial charges? What would be the MO diagram that corresponds to $S = 3/2$ in this specific ligand field of 11 ligand atoms?

(2) Not all values of the bonding analysis are self-explanatory. For example, a DI of 0.78 should be classified.

(3) It may be instructive to compare this metallocene with other highly reduced species, such as CO and CN complexes with high electron count.

Minor remarks:

(4) Complex "1" should be named at least once.

(5) "Comparable" should be used only to express that two things have common ground that makes a comparison meaningful, but not as a non-specific synonym for "similar."

(6) The heading "Observation of interatomic interactions" is somewhat inaccurate, since the interactions (taking into account experimental data) were nevertheless only calculated.

Point-by-point response to the reviewers' comments

Reviewer #1 (Remarks to the Author):

This manuscript presents a comprehensive investigation of a 21-electron cobalt complex stabilized by a constrained geometry configuration ligand. The authors have extensively characterized the compound, performed detailed theoretical calculations, and studied its reduction reactions to confirm the presence of Co-N bonds, thus proving the 21-electron structure of the central metal. The characterization analysis is convincing, and the results are sufficiently innovative. However, I have some reservations regarding the compound's significance as it may be an isolated case. Therefore, I recommend this manuscript for publication in Nat. Comm. after the following issues are properly addressed:

The authors mention their unsuccessful attempt to synthesize a similar complex with a Ni center. Have they explored Fe and Mn as well? It would be valuable to provide experimental information demonstrating that this ligand design strategy is not limited to cobalt, considering the relative ease of such synthesis attempts.

Response: Thank you very much for this suggestion. We included the synthesis and characterization of [Mn(CpNCp)] complex. Regarding the Fe analogue, we are preparing another report that focuses on the chemistry of [Fe(CpNCp)] complexes.

The authors propose that the compound's stability arises from the lower energy of the three half-filled orbitals formed by N coordination. However, this statement raises questions, as it implies that cobaltocene in pyridine solution would exhibit the same electronic structure if energetically favorable.

Response: We proposed that the formation of the three half-filled orbitals is one of the reasons for the stability of **1**. However, we think the chelate effect of the pincer ligand is essential to form and stabilize the formal 21-electron complex. To the best of our knowledge, the coordination of two-electron donor to cobaltocene without dissociation of a Cp ligand is not known. Cp ligand in cobaltocene is known to be substituted under forcing conditions (e.g. <https://onlinelibrary.wiley.com/doi/abs/10.1002/9780470132593.ch71>). We examined if pyridine could coordinate to cobaltocene in a neat pyridine solution. cw-EPR (4.2 K in 6:1 pyridine:C₆D₆) and ¹H NMR (-40, 25, 80 °C in 6:1 pyridine:C₆D₆ with an external standard for Evans method) measurements did not show evidence that supports the formation of S = 3/2 species similar to **1**.

Reviewer #2 (Remarks to the Author):

First, I must note that I am a synthetic chemist with an interest in the use of stable metallocenes (ferrocene..) for ligand synthesis/catalysis. Thus many of the details and techniques used in this manuscript are outside of my area of expertise. This notwithstanding, it appears that the work is thorough and very well done. I'm assessing this principally on the novelty of the work, and in this context the claim for a formal 21-electron metallocene derivative is significant, and to my (perhaps limited) knowledge is established beyond doubt. Of particular note is the oxidised cobaltocenium compound **3** which does not contain the C-N bond (i.e. just 18 electron), and the calculations comparing **1'** with **1** and the experimental verification of the 3/2 spin state. I'm less convinced about the possible impact of what appears to be an organometallic oddity on the wider fields of catalysis and materials, but that is for the future, and on the basis of the novelty of the result I am happy to recommend publication.

Minor point/suggestions.

Include solvent label in the ¹H NMR (in addition to THF)?

Response: In Fig. 2a, there are three solvent signals, one for C₆D₅H and two for THF. We labeled C₆D₅H signal in addition to the THF signals.

As THF is present, is the microanalysis data correct?

Response: According to the integration of ¹H NMR spectrum used in Fig. 2a, THF content in this specific batch of **1** is about 3.5%, even though it is not recorded using a quantitative NMR method. If we include this in the theoretical CHN value, the value becomes C:69.84%, H:5.23%, N: 4.75%. Our experimental value is within the usually acceptable error (±0.4%) of this value or the value without considering 3.5% THF.

Figure 5. Change orbital representations so they are aligned the same as the structures (pyridine left).

Response: Thank you for the suggestion. We changed it according to your suggestion.

Reviewer #3 (Remarks to the Author):

The authors present a new metallocene that for the first time has a nominal electron count as high as 21, significantly exceeding the known 18-electron rule. That such a bonding situation could be stabilized is remarkable, since the additional electrons occupy antibonding states. To achieve this, they used a pyridine-based pincer ligand that provides an addition

coordinating N atom. Given the authors' experimental results on their attempts to make similar complexes and the very specific electronic situation they uncovered through DFT calculations (competition to N not coordinating), it is not very likely that the concept can be applied to many other cases. Nonetheless, this is an important contribution to the largely established, yet still explored, field of metallocenes. The used combination of experimental techniques and quantum chemical calculations provides a sound picture of the nature of the new compound. The manuscript was developed with care and has a high formal standard. Still the manuscript could be improved in few aspects.

(1) Can the results of the extended bonding analysis be condensed into a (simplified) picture of chemical bonding that can be adopted in textbooks? Where are the electrons? What are the calculated partial charges? What would be the MO diagram that corresponds to $S = 3/2$ in this specific ligand field of 11 ligand atoms?

Response: Thank you for the suggestion. We considered improving Fig.5 to make it more informative, however, it is difficult to draw a simplified picture as the key bonding interactions in complex **1** involve half-filled antibonding orbitals. MO diagram and orbitals in Fig.5 show key MOs and where key electrons are, and we think the figure is simple enough to explain the key bonding situation in **1** to a non-expert. The calculated partial charges are included in Table S3, and a brief discussion about the charge is included in the text. A full MO diagram and a full list of Kohn–Sham quasi-restricted orbitals are provided in the supplementary information.

(2) Not all values of the bonding analysis are self-explanatory. For example, a DI of 0.78 should be classified.

Response: Thank you for pointing out this. We added a new reference (ref. 53) where the reader can find further explanations on the concept of DI as well as a comprehensive list of DIs for simple molecules.

(3) It may be instructive to compare this metallocene with other highly reduced species, such as CO and CN complexes with high electron count.

Response: Thank you very much for this suggestion. We included a comparison between our electronic situation and that of formal 20-electron $M(\text{CO})_8$ ($M = \text{Zr}, \text{Hf}$) and $[\text{M}(\text{CO})_8]^-$ ($M = \text{Sc}, \text{Y}, \text{La}$) complexes reported by G. Frenking group.

<https://onlinelibrary.wiley.com/doi/10.1002/anie.201802590>,

<https://chemistry-europe.onlinelibrary.wiley.com/doi/full/10.1002/chem.201905552>

Minor remarks:

(4) Complex "1" should be named at least once.

Response: We named **1** as 2,6-bis(methylenecyclopentadienyl)pyridinecobalt.

(5) "Comparable" should be used only to express that two things have common ground that makes a comparison meaningful, but not as a non-specific synonym for "similar."

Response: We replaced "comparable" with "similar".

(6) The heading "Observation of interatomic interactions" is somewhat inaccurate, since the interactions (taking into account experimental data) were nevertheless only calculated.

Response: We replaced "Observation" with "Examination".

Reviewers' Comments:

Reviewer #1:

Remarks to the Author:

I have noticed that the author has revised and polished the manuscript for each question, and this version is convincing and suitable for publication.

Reviewer #3:

Remarks to the Author:

The authors have revised their manuscript, including additional data and explanations. The issues raised by this reviewer were almost all resolved. The current version of the manuscript is suitable and recommended for publication in Nat. Commun.

Point-by-point response to the reviewers' comments (2nd round)

Reviewer #1 (Remarks to the Author):

I have noticed that the author has revised and polished the manuscript for each question, and this version is convincing and suitable for publication.

Response: Thank you very much for reviewing our manuscript.

Reviewer #3 (Remarks to the Author):

The authors have revised their manuscript, including additional data and explanations. The issues raised by this reviewer were almost all resolved. The current version of the manuscript is suitable and recommended for publication in Nat. Commun.

Response: Thank you very much for reviewing our manuscript.